# Clinical significance of central systolic blood pressure in LV diastolic dysfunction and CV mortality

Sun Ryoung Choi[1,2], Young-Ki Lee[2,3]*, Hayne Cho Park[2,3], Do Hyoung Kim[2,3], Ajin Cho[2,3], Min-Kyung Kang[4], Seonghoon Choi[4]

1 Department of Internal Medicine, Hallym University Dongtan Sacred Heart Hospital, Dongtan, Korea,
2 Hallym University, Kidney Research Institute, Seoul, Korea, 3 Department of Internal Medicine, Hallym University Kangnam Sacred Heart Hospital, Seoul, Korea, 4 Division of Cardiology, Department of Internal Medicine, Hallym University Kangnam Sacred Heart Hospital, Seoul, Korea

* km2071@naver.com

## Abstract

**Data Availability Statement:** All relevant data are within the paper and its Supporting Information files.

### Background and aims

Cardiovascular (CV) disease is the major cause of death in patients with end-stage kidney disease (ESKD). Left ventricular (LV) diastolic dysfunction reflects LV pressure overload and is common in patients with ESKD. Recently, there have been studies on the usefulness of central blood pressure (BP); however, the relationship between central BP and LV diastolic dysfunction is not clear in dialysis patients with preserved systolic function. The purpose of this study was to investigate the clinical implication of central BP on LV diastolic dysfunction and CV mortality in the ESKD patients with preserved LV systolic function.

### Methods

This prospective observational cohort study investigated the effect of LV diastolic dysfunction on CV mortality in the ESKD patients with preserved systolic function using echocardiography. Vascular calcification was evaluated using the abdominal aortic calcification score, and aortic stiffness was determined by measuring pulse wave velocity (PWV). The predictors of CV mortality were analyzed using Cox proportional hazard analysis.

### Results

The subjects were comprised of 61 patients, with an average age of 54 years, 20 males (32.8%), and 25 diabetics (41.0%). There were 39 patients on hemodialysis and 22 patients receiving peritoneal dialysis treatment. During the mean follow-up period of 79.3 months, 16 patients (26.2%) died, and 7 patients (11.4%) died of CV events. The central systolic BP and diabetes were independent risk factors for LV diastolic dysfunction. In addition, patients with LV diastolic dysfunction had an increased CV mortality. When left atrial volume index and PWV were adjusted, the E/E' ratio was found to be a predictor of CV mortality.

**Funding:** This research was supported by Hallym University Research Fund 2020 (HURF-2020-23).

**Competing interests:** The authors declare no conflicts of interest

## Conclusions

Central systolic BP and diabetes were found to be significant risk factors for LV diastolic dysfunction. LV diastolic dysfunction can independently predict CV mortality in dialysis patients with preserved LV systolic function.

## Introduction

Left ventricular (LV) diastolic dysfunction is frequently observed in dialysis patients and is associated with heart failure and higher cardiovascular (CV) mortality. The prevalence and severity of LV diastolic dysfunction gradually increase as kidney function deteriorates, and it occurs in approximately 54.3% of patients on hemodialysis (HD), even in those without apparent symptoms [1]. The known risk factors of LV diastolic dysfunction are age, hypertension, diabetes, ischemic heart disease, and LV hypertrophy (LVH). These factors are often associated with the development of myocardial fibrosis and decreased ventricular compliance [2]. LVH is frequently complicated, accompanied by myocardial fibrosis and LV dysfunction [3]. These structural changes predispose the cardiovascular system of patients with ESKD to LV dysfunction. Of note, regression of LVH in patients with end-stage kidney disease (ESKD) was found to have a favorable and independent effect on CV mortality [4, 5].

The most important cause of mortality in patients with ESKD is cardiovascular disease (CVD); 50% of dialysis patients die of CVD [6, 7]. CVD-related risk factors and adaptive alterations include LVH and LV dilatation with concomitant LV systolic and diastolic dysfunction [8]. The pathophysiologic adjustment of ESKD includes endothelial dysfunction, volume or pressure overload, and LV systolic and diastolic dysfunction [9]. These alterations are also thought to play an important role in CV morbidity and mortality [8]. Kang et al. stated that LV diastolic dysfunction is an independent predictor of CV events in patients with ESKD with preserved systolic function [10]. Furthermore, the tissue Doppler imaging (TDI) parameters may reflect the impairment of LV pressure overload and diastolic dysfunction defined by TDI, and is an independent predictor of mortality in patients with ESKD with preserved LV systolic function [11].

Central blood pressure (BP) exhibits a strong association with LV filling pressure and LV diastolic dysfunction [12], and the measurement of central BP is better related to future CV events compared to brachial BP [13, 14]. However, there is limited data evaluating the value of predicting CV risk via central BP measurement in patients with ESKD [15]. Furthermore, the clinical implication of central BP and LV diastolic dysfunction on CVD mortality remains unclear. Therefore, the purpose of this study was to determine the impact of LV diastolic dysfunction on CV mortality in patients with ESKD with preserved systolic function and to analyze the relationship between central BP and LV diastolic dysfunction.

## Materials and methods

### Study population

This single-center prospective observational study enrolled patients who had undergone HD or peritoneal dialysis (PD) from April 1, 2011, to December 31, 2013. In addition, patients were followed to assess CV mortality until May 31, 2020. This study was conducted at Hallym University Kangnam Sacred Heart Hospital (Seoul, Korea). It involved patients with ESKD who were older than 20 years and who had been undergoing HD or PD for more than 3

months. Patients with CVD within the last 3 months; ejection fraction (EF) of less than 50% or mitral regurgitation (MR) grade II or higher; or history of active infection, malignancy, chronic lung disease, or rheumatoid disease were excluded from the study. This study was approved by the Institutional Review Board of Hallym University Kangnam Sacred Hospital (IRB No: 2010-08-061). Informed written consent was obtained from each patient before participation in the study.

## Clinical and biochemical parameters

CVD was defined as a history of coronary, arrhythmia, peripheral vascular disease or cerebro-vascular disease. Hypoalbuminemia was defined as a serum albumin level of less than 4.0 g/dL and a hemoglobin level of less than 10 g/dL. Requiring treatment with an erythropoiesis stimulation agent was defined as anemia. Baseline parameters including demographic, laboratory, and dialysis-related data were collected during study enrollment. Blood samples were obtained before the midweek dialysis session of HD patients and 2 hours after the first PD exchange with 1.5% dextrose dialysate of PD patients using standard techniques. The following data were measured: levels of hemoglobin, blood urea nitrogen, serum creatinine, calcium, phosphorus, albumin, total cholesterol, triglyceride, intact parathyroid hormone, alkaline phosphatase, and high-sensitivity C-reactive protein. Kt/V was calculated using the logarithmic estimate of the Daugirdas method [16].

## Estimation of central blood pressure

The central BP, pulse pressure, and augmentation index (AI) were measured by pulse waves detected in the radial artery pressure waveforms using the HEM-9000AI (Omron Healthcare, Kyoto, Japan) [17]. The augmentation of central BP is a manifestation of early wave reflection and is the boost of pressure from the first systolic shoulder to the systolic pressure peak. Augmentation is calculated as the difference between the second and first systolic shoulder of the central pressure wave curve, and AI is expressed as the percentage of augmentation from PP. Because AI is influenced in an inverse and linear manner by heart rate according to Wilkinson et al. [18], it was normalized for a heart rate of 75 bpm (AI@75). For HD patients, central BP, pulse pressure, and AI were measured before and after dialysis.

## Assessment of brachial-ankle Pulse Wave Velocity (PWV), Ankle Brachial Index (ABI) and Abdominal Aortic Calcification (AAC)

The brachial-ankle PWV was obtained using a waveform analyzer (VP-2000; Colin Co Ltd, Komaki, Japan) as previously described [19]. ABI was measured according to the recommended method [20]. The AAC of the subjects was measured at the start of the study. The AAC score was calculated using a previously validated method by Kauppila et al. [21]. Lateral lumbar radiographs were obtained using a standard radiographic equipment. The severity of the anterior and posterior aortic calcification was graded individually on a 0–3 scale for each lumbar segment (L1–L4), and the results were summarized to develop the AAC score (range 0–24).

## Echocardiography

Transthoracic echocardiograms were performed at baseline on the non-dialysis day for HD patients or during the dwell phase of PD patients and were obtained by fundamental imaging (two-dimensional), M-mode, and tissue Doppler imaging (TDI)) using a 2.5-MHz transducer and a commercial ultrasound system (Vivid 7, GE Vingmed Ultrasound AS, Horten, Norway).

Chamber dimension, wall thickness, and LV ejection fraction (EF) were measured (M-mode), and the mitral annular velocities were obtained by tissue Doppler imaging (TDI). The left atrial (LA) dimension was determined from M-mode echocardiograms using a leading-edge to leading-edge technique, measuring the maximal distance between the posterior aortic root wall and the posterior LA wall at the end systole [22]. LA enlargement was defined as a larger left atrial volume index (LAVI) of 34 mm/m$^2$ in both sexes [23, 24]. We measured the early trans-mitral flow velocity (E), early mitral annular velocity (E'), and calculated the E/E' ratio. The E/E´ ratio reflects the mean LV diastolic pressure, and an E/E' ratio of greater than15 indicates LV diastolic dysfunction [25]. Based on the report, LV diastolic dysfunction was defined as an E/E' ratio greater than 15. LV mass was calculated using the formula LV mass (g) = 0.8 × [1.04 × (left ventricle internal dimension diastole (LVIDd) + posterior wall (PW) + inter-ventricular septum diastole (IVSd))$^3$ - (LVIDd)]$^3$ − 0.6, and LV mass index (LVMI) was calculated as LV mass (in g)/ body surface area. According to the current American and European guidelines, LVH was defined as an LVMI of at least 115 g/m$^2$ in men and 95 g/m$^2$ in women [26]. LV systolic function was assessed by calculating the EF using a modified Simpson's method, and LV systolic dysfunction was defined as EF <50% [27]. Echocardiography was performed by an experienced specialist who was blinded to patient information.

## Statistical analysis

Statistical analyses were performed using the SPSS software version 21 (IBM Corp., Armonk, NY, USA). Summary statistics are expressed as means ±, standard deviations, or medians for continuous variables, and as frequencies or percentages for categorical variables. Continuous variables were analyzed using the Mann–Whitney U test. Spearman's correlation coefficient analysis was used to evaluate the linear relationship between two continuous variables. Categorical variables were compared using the chi-square test or Fisher's exact test, as appropriate. The patients were divided into two groups according to their E/E' ratio. Univariate and multivariate logistic regression analyses were performed to identify the independent risk factors of LV diastolic dysfunction. The Kaplan–Meier method for survival analysis, and log-rank test were used to compare the survival rate differences between patients with and without LV diastolic dysfunction. Cox proportional- hazards regression models were constructed to evaluate the influence on CV mortality. A p-value of less than 0.05 was considered statistically significant.

## Results

### Baseline characteristics

Of the 75 patients, only 61 participated in this study, excluding 14 patients with an EF of less than 50% or an MR grade of II or higher. The study subjects were divided into two categories: with or without LV diastolic dysfunction based on the E/E' ratio of 15. The mean age of the 61 patients was 54.1±12.5 (range, 29–81 years), and 20 patients (32.8%) were men (Table 1). The underlying cause of ESKD was diabetes in 25 patients (41.0%). Among all patients, 25 (41.0%) had a past history of CVD. The mean dialysis vintage was 50.8 (range, 4–249) months. Overall, 67.2% of the patients had LVH and 44.3% had an E/E' ratio of 15 or higher. When compared based on the E/E' ratio of 15, the central systolic BP and pulse pressure were significantly higher in the higher E/E' ratio group than in the lower E/E' ratio group ($p<0.005$). PWV and AAC scores were significantly increased in patients with a higher E/E' ratio ($p<0.05$), but the ABI did not differ between the two groups. LA dimension and LAVI were significantly higher in the higher E/E' ratio than in the lower E/E' ratio group ($p<0.05$). There was no difference in

**Table 1. Baseline characteristics of the study population according to E/E' ratio at baseline.**

| | All | E/E' ratio ≤15 | E/E' ratio >15 | *p* value |
|---|---|---|---|---|
| | (n = 61) | (n = 34) | (n = 27) | |
| Demographic data | | | | |
| Age, years | 54.1±12.5 | 52.3±13.5 | 56.5±10.7 | 0.180 |
| Male, n (%) | 20 (32.8) | 12 (35.3) | 8 (29.6) | 0.363 |
| HD, n (%) | 39 (63.9) | 19(55.9) | 20(74.1) | 0.114 |
| Comorbidities, n (%) | | | | |
| Diabetes Mellitus | 25 (41.0) | 7 (20.6) | 18 (66.7) | 0.001 |
| Cardiovascular disease | 25 (41.0) | 11 (32.4) | 14 (51.9) | 0.104 |
| Dialysis vintage, years | 4.1±3.9 | 3.6±2.9 | 4.9±4.9 | 0.206 |
| BMI, kg/m$^2$ | 22.6±3.3 | 22.9±3.4 | 22.3±3.1 | 0.439 |
| Kt/V | 1.52±0.28 | 1.57±0.25 | 1.46±0.31 | 0.119 |
| Central systolic BP, mmHg | 150.2±26.6 | 141.7±25.8 | 160.9±23.9 | 0.004 |
| Central diastolic BP, mmHg | 79.4±13.9 | 81.2±11.4 | 77.0±16.7 | 0.255 |
| Central pulse pressure, mmHg | 70.5±22.8 | 60.8±20.6 | 82.8±19.6 | 0.000 |
| AI, % | 80.4±17.2 | 77.1±17.6 | 84.6±12.6 | 0.078 |
| AI@75, % | 80.7±16.5 | 77.8±17.6 | 84.4±14.4 | 0.121 |
| PWV, m/sec | 1.68±0.4 | 1.58±0.37 | 1.82±0.42 | 0.018 |
| ABI | 1.47±1.20 | 1.18±0.10 | 1.21±0.12 | 0.236 |
| Presence of AAC, n (%) | 33 (54.1) | 15(44.1) | 18(66.7) | 0.121 |
| AAC score | 3.4±4.7 | 2.1±3.4 | 5.0±5.6 | 0.020 |
| Laboratory data | | | | |
| Hemoglobin, g/dL | 10.1±1.0 | 10.1±0.9 | 10.2±1.0 | 0.605 |
| hs-CRP, mg/L | 2.7±4.6 | 2.8±5.4 | 2.6±3.1 | 0.868 |
| Albumin, g/dL | 3.9±0.4 | 3.9±0.4 | 3.9±0.4 | 0.487 |
| Total cholesterol, mg/dL | 151.9±35.9 | 147.6±32.9 | 157.6±39.6 | 0.270 |
| iPTH, pg/mL | 187.4±186.6 | 170.3±160.7 | 210.5±217.9 | 0.391 |
| Calcium, mg/dL | 8.6±0.8 | 8.6±0.8 | 8.7±0.8 | 0.969 |
| Phosphorus, mg/dL | 4.6±1.5 | 4.6±1.7 | 4.9±1.3 | 0.959 |
| Echocardiography | | | | |
| LAD, mm | 54.9±23.8 | 49.2±23.5 | 62.1±22.6 | 0.035 |
| LAVI, ml/m$^2$ | 33.6±13.1 | 29.9±12.8 | 38.2±12.2 | 0.012 |
| LVMI, g | 128.6±41.6 | 121.0±40.9 | 138.2±41.2 | 0.110 |
| LVH, n (%) | 41(67.2%) | 22 (64.7) | 19 (70.3) | 0.425 |
| EF, % | 64.4±6.6 | 66.4±6.2 | 61.9±6.5 | 0.009 |

AI, augmentation index; AI@75, augmentation index and normalized for heart rate equal 75 beats/minute, ABI, ankle brachial index; AAC, abdominal aortic calcification; BMI, body mass index; BP, blood pressure; LAD, left atrial dimension; LAVI, left atrial volume index; LVMI, Left ventricular mass index; LVH, left ventricular hypertrophy; E, early diastolic mitral inflow velocity; E', early diastolic mitral annular velocity; EF, ejection fraction; hs-CRP, high-sensitivity C-reactive protein; iPTH, intact parathyroid hormone.

the ratio of LVH between the two groups. Serum levels of hemoglobin and albumin did not differ between the two groups.

## LV diastolic dysfunction and predictors of CV mortality

The mean follow-up duration was 79.3±40.0 months. A total of 16 (26.2%) patients died during the follow-up period. Seven (43.8%) of the 16 deaths were due to CVD. Three out of seven died from myocardial infarction, one from sudden cardiac death, and the other three from

stroke. Of all the deaths, 9 were because of infections such as pneumonia, peritonitis, and sepsis. All-cause mortality was not significantly different between patients with or without LV diastolic dysfunction ($p$ = 0.283). However, Kaplan-Meier survival analysis showed that the cumulative incidence rates of CV mortality were significantly higher in patients with LV diastolic dysfunction (log-rank test, $p$ = 0.021, Fig 1). The E/E' ratio was a significant risk factor, as revealed by multivariate Cox regression proportional hazard analysis for CV mortality (hazard ratio [HR] = 1.150, 95% confidence interval [CI] = 1.010–1.309, $p$ = 0.034; Table 2).

### Independent risk factor for LV diastolic dysfunction

In univariate logistic regression analysis, the risk factors for LV diastolic dysfunction were identified as diabetes, central systolic BP, PWV, and AAC score. However, multivariate logistic regression analysis showed that central systolic BP (HR = 1.034, 95% CI = 1.002–1.068, $p$ = 0.036) and diabetes (HR = 9.373, 95% CI = 2.382–36.883, $p$ = 0.001) were significant independent risk factors for LV diastolic dysfunction (Table 3). Central systolic BP showed a significant positive correlation with the E/E' ratio (r = 0.441, $p$ = 0.000; Fig 2A) and LAVI (r = 0.394, $p$ = 0.005, Fig 2B). In addition, PWV (r = 0.510, p = 0.000) and AAC score (r = 0.286, $p$ = 0.025) had a significant positive correlation with central systolic BP. However, central diastolic BP had no correlation with the E/E' ratio, LAVI, PWV, and AAC score.

## Discussion

This study revealed that LV diastolic dysfunction increased CV mortality in patients receiving HD and PD. The E/E' ratio was a significant factor for CV mortality. Central systolic BP was a significant independent risk factor for LV diastolic dysfunction. Furthermore, central systolic BP had a significant positive correlation with E/E' ratio and LAVI.

Elevated LV filling pressure is the main physiological finding in LV diastolic dysfunction and is associated with symptom onset [2]. The E/E' ratio and LAVI have been shown to reliably assess LV diastolic dysfunction in dialysis patients as well as in the general population [28]. LV diastolic dysfunction is an independent predictor of CV morbidity and mortality in the general population and in dialysis patients [10]. Recently, Kang et al. showed the implications of LV diastolic dysfunction on CV events in incident dialysis patients with preserved LV systolic function [10]. The difference between the aforementioned study and the present study is that we studied LV diastolic dysfunction on CV mortality in maintenance HD and PD patients. Moreover, Yu et al. suggested that LV diastolic dysfunction as evaluated by TDI is an independent predictor of hospitalization and all-cause mortality in dialysis patients with preserved LV systolic function [11]. In that study, 9.1% of all patients had diabetes and only 4% had a history of CVD. However, 41% had diabetes and 41% had a history of CVD in our study. For these reasons, it is judged that there is a difference in the results. Moreover, there was no mention of the effect of LV diastolic dysfunction on CV mortality in the aforementioned study.

Subherwal reported that central BP is independently associated with LV filling pressure and LV diastolic dysfunction in the general population [12]. Summarizing the research literature reported to date, central BP had a significant association with LV diastolic dysfunction, which has been shown to be a predictor of CV events and mortality in patients with ESKD. However, there has been no study between central BP and LV diastolic dysfunction on CV mortality in ESKD patients with preserved LV systolic function.

BP has been traditionally measured in the peripheral arteries. However, there is increasing evidence that central BP might be superior to peripheral BP in the prediction of CVD events [29]. The Anglo-Scandinavian Cardiac Outcomes Trial-Blood Pressure Lowering Arm and

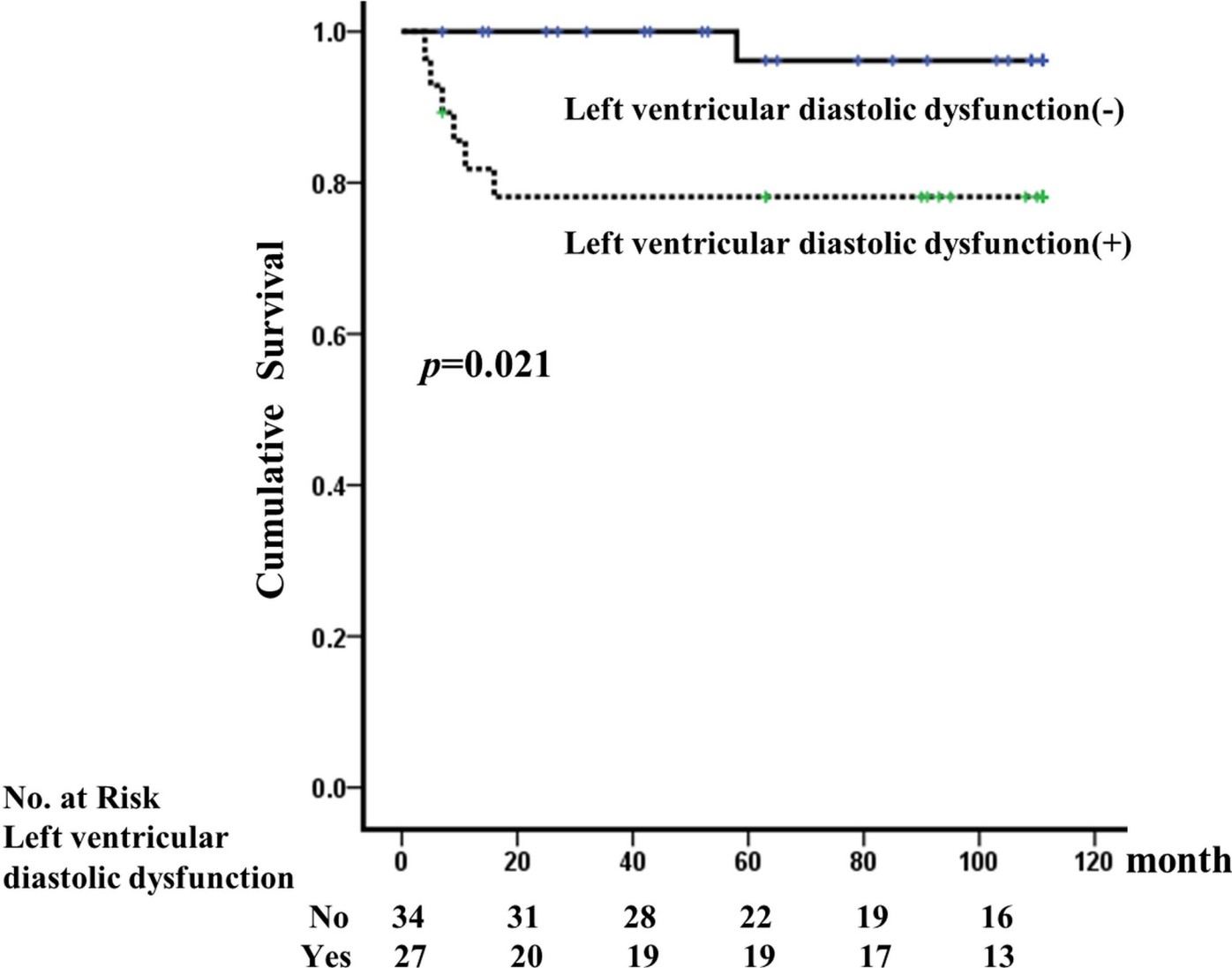

**Fig 1. Kaplan-Meier survival curves for CV mortality according to LV diastolic dysfunction.** CV mortality was significantly higher in patients with LV diastolic dysfunction (log-rank test, *p* = 0.021).

Conduit Artery Function Evaluation (CAFE) demonstrated the impact of central BP on the risk of CVD on the general population [30, 31]. However, clinical implications of central BP are rare in ESKD patients with preserved LV function. Increased aortic afterload by central BP results in LV diastolic dysfunction, characterized by impaired LV relaxation and increased LV end-diastolic pressure. Consequently, elevated central BP may contribute to LV diastolic dysfunction [12]. In a study involving 180 people who received dialysis treatment, brachial BP had no predictive value for mortality after adjustment. However, central BP was more powerful than brachial BP in the prediction of overall mortality [32]. However, the value of central BP measurements in predicting CVD outcome has been studied mostly in the general population with hypertension, but not in detail in patients with ESKD [33]. Mahboob et al. reported that elevated central BP is associated with a higher risk of CVD outcomes in patients with CKD (eGFR between 20 and 70 ml/min per 1.73 m$^2$) [15]. In the present study involving chronic dialysis patients, central systolic BP had a strong positive correlation with the E/E' ratio and

**Table 2. Cox regression analysis of proportional hazard for CV mortality.**

| | Univariate | | | Multivariate | | |
|---|---|---|---|---|---|---|
| | Hazard ratio | 95% CI | *p* value | Hazard ratio | 95% CI | *p* value |
| Age, year | 1.017 | 0.960~1.078 | 0.570 | | | |
| Male | 1.624 | 0.363~7.258 | 0.526 | | | |
| HD over PD | 1.372 | 0.266~7.071 | 0.706 | | | |
| History of CVD | 1.959 | 0.438~8.755 | 0.379 | | | |
| BMI, kg/m$^2$ | 1.129 | 0.931~1.370 | 0.218 | | | |
| Central systolic BP, mmHg | 1.007 | 0.979~1.036 | 0.613 | | | |
| Central diastolic BP, mmHg | 0.964 | 0.893~1.041 | 0.347 | | | |
| PWV, m/sec | 1.001 | 1.000~1.003 | 0.078 | 1.001 | 0.998~1.003 | 0.638 |
| Presence of AAC | 2.141 | 0.415~11.036 | 0.363 | | | |
| Hypoalbuminemia | 1.196 | 0.268~5.351 | 0.814 | | | |
| Anemia, g/dL | 1.081 | 0.242~4.830 | 0.919 | | | |
| hs-CRP, mg/L | 1.007 | 0.866~1.170 | 0.930 | | | |
| LAVI, ml/m$^2$ | 1.053 | 1.001~1.108 | 0.046 | 1.038 | 0.977~1.102 | 0.226 |
| E/E' ratio | 1.189 | 1.068~1.324 | 0.002 | 1.150 | 1.010~1.309 | 0.034 |

AAC, abdominal aortic calcification; BP, blood pressure; CVD, cardiovascular disease; BMI, body mass index; LAVI, Left atrial volume index; PWV, pulse wave velocity; PP, pulse pressure; HD, hemodialysis; PD, peritoneal dialysis; E, early diastolic mitral inflow velocity; E', early diastolic mitral annular velocity.

PWV. In addition, central systolic BP was a significant risk factor for LV diastolic dysfunction in ESKD patients with preserved LV systolic function. However, Kanako et al. insisted measuring central BP was useful in detecting minute changes in arteries, which would not change the

**Table 3. Univariate and multivariate logistic regression analysis for LV diastolic dysfunction.**

| | | Univariate | | | Multivariate | |
|---|---|---|---|---|---|---|
| | Beta | Hazard ratio (95% CI) | P value | Beta | Hazard ratio (95% CI) | *P* value |
| Age, year | 0.022 | 1.023(0.980~1.067) | 0.304 | | | |
| Male | 0.000 | 1.000(0.350~2.858) | 1.000 | | | |
| DM | 2.043 | 7.714(2.433~24.456) | 0.001 | 2.238 | 9.373(2.382~36.883) | 0.001 |
| HD versus PD | 0.714 | 2.042(0.707~5.895) | 0.187 | | | |
| Central systolic BP | 0.031 | 1.031(1.008~1.055) | 0.007 | 0.034 | 1.034(1.002~1.068) | 0.036 |
| Central diastolic BP | -0.015 | 0.985(0.949~1.023) | 0.434 | | | |
| BMI, kg/m$^2$ | -0.075 | 0.928(0.791~1.089) | 0.360 | | | |
| Dialysis vintage, years | -0.016 | 0.985(0.856~1.132) | 0.827 | | | |
| PWV, m/sec | 0.002 | 1.002(1.000~1.003) | 0.027 | -0.001 | 0.999(0.997~1.001) | 0.490 |
| AAC score | 0.148 | 1.160(1.023~1.314) | 0.020 | 0.155 | 1.168(0.989~1.378) | 0.067 |
| Hypoalbuminema | -0.223 | 0.800(0.290~2.205) | 0.666 | | | |
| Anemia, g/dL | 0.018 | 1.018(0.364~2.845) | 0.973 | | | |
| Calcium, mg/dL | 0.066 | 1.068(0.572~1.994) | 0.836 | | | |
| Phosphorus, mg/dL | 0.034 | 1.034(0.739~1.447) | 0.844 | | | |
| iPTH, pg/mL | 0.001 | 1.001(0.999~1.004) | 0.333 | | | |
| hs-CRP, mg/L | -0.015 | 0.985(0.879~1.103) | 0.792 | | | |

AAC, abdominal aortic calcification; BMI, body mass index; DM, diabetes mellitus; SBP, systolic blood pressure; iPTH, intact parathyroid hormone; PWV, pulse wave velocity; HD, hemodialysis; PD, peritoneal dialysis.

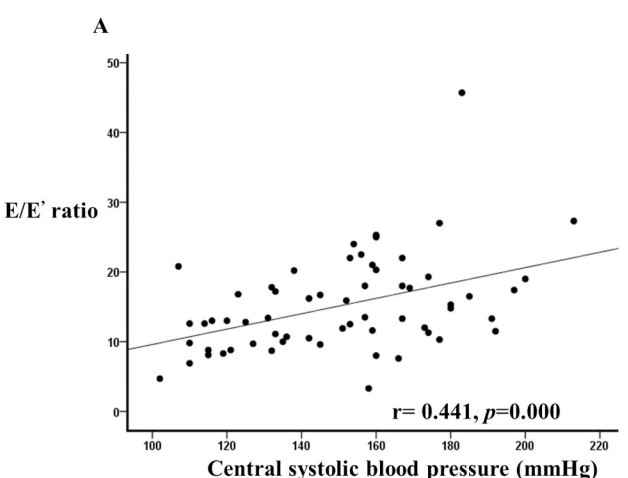

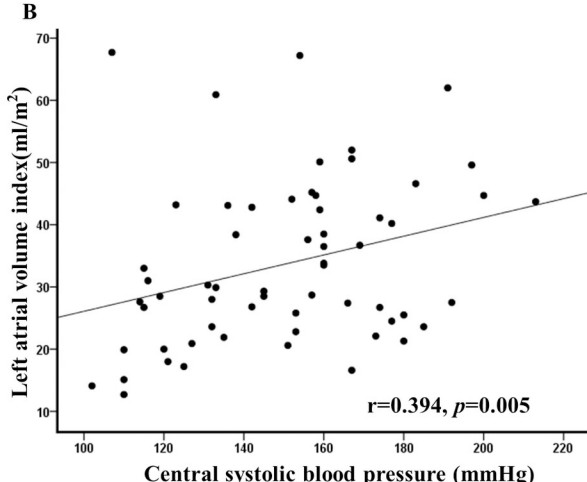

**Fig 2.  A) Central systolic BP had positive correlation with E/E' ratio (r = 0.441, *p* <0.001), B) Central systolic BP and LAVI had positive correlation (r = 0.394, *p* = 0.005).**

brachial BP [34]. In contrast, Ichihara et al. [35] observed a significant reduction in PWV with treatment in dialysis patients. Thus, it seems that the arteries in dialysis patients can respond to adequate treatment on the basis of this study. Moreover, Mahboob et al. [15] reported that the measurement of central BP may be useful in patients with CKD to determine their risk of CVD. Therefore, measurement of central BP may be suitable even for dialysis patients despite advanced arterial disease due to chronic inflammation and advanced arterial calcification [34].

Although there are some reports that hypertension affects CV mortality in dialysis patients [36, 37], there are few studies on central BP and CV mortality in dialysis patients. Since the number of subjects was small and the observation period was not long enough, our study might not show significant association between central systolic BP and CV mortality. In addition, numerous risk factors are involved in the pathogenesis of CVD in dialysis patients. There are well-known traditional risk factors such as diabetes and hypertension as well as non-traditional risk factors such as chronic volume overload, anemia, inflammation, oxidative stress, chronic kidney disease–mineral bone disorder and other aspects of the 'uraemic milieu' [38]. Further research should be performed to reveal the relationship between central systolic BP and CV outcome.

Diabetes is one of the most important metabolic conditions responsible for LV diastolic dysfunction. Although the exact cause has not yet been identified, diabetic cardiomyopathy has been proposed recently. Clinically, it is well known that LV diastolic function deteriorates with advancing age despite preserved LV systolic function. In particular, impaired glucose tolerance could contribute to the early deterioration of LV diastolic function among middle aged apparently healthy subjects [39]. Similarly, in this study, patients with diabetes had significantly more LV diastolic dysfunction. Furthermore, DM was identified as a strong risk factor for LV diastolic dysfunction with HR of 9.373 (*p* = 0.001). However, we think further research is warranted to generalize our results.

Tullio et al. reported that increased BMI (>25 kg/m$^2$) was associated with worse LV diastolic function independent of LV mass and associated risk factors in general population [40]. However, unlike the general population, among HD patients BMI is related to mortality inversely. The explanation for this interesting paradox is unknown [41]. Agarwal et al. shows a significant and inverse relationship of BMI with unadjusted LVMI, but multivariate analysis

removed the statistical association of BMI with LVMI [42]. We did not find any studies on BMI and LV diastolic dysfunction in dialysis patients. In our study, BMI was not a significant risk factor for LV diastolic dysfunction in a univariate analysis ($p = 0.360$). Furthermore, overweight and obesity with a BMI value of 25 or higher were not a significant risk factor for LV diastolic dysfunction in this study ($p = 0.465$). The authors believe that studies on BMI, LV diastolic dysfunction, and CV mortality in dialysis patients are of great value.

The prevention and treatment of LV diastolic dysfunction in patients with ESKD may alleviate or regress LVH. Since central systolic BP was found to be a significant risk factor for LV diastolic dysfunction in this study, it is believed that CV mortality may be reduced by meticulous monitoring and control of central systolic BP.

This study has several limitations. First, this was a single-center study with a small sample size. Second, an observational study may only provide an associative link but not a causative link; therefore, we cannot rule out the possibility of unmeasured confounding factors that can influence the implications of CV outcomes. Third, if echocardiography was performed at least every other year, more accurate and clear results could have been obtained. Lastly, we did not sequentially examine the patients for the presence of CV outcomes or risk factors such as nutritional, inflammatory, and volume status during the follow-up period, which would have yielded more informative results.

## Conclusion

LV diastolic dysfunction can independently predict CV mortality in dialysis patients with preserved LV systolic function. The E/E' ratio was an independent predictor of CV mortality in dialysis patients. In addition, central systolic BP was a significant independent risk factor for LV diastolic dysfunction.

## Supporting information

**S1 File.**
(SAV)

## Author Contributions

**Conceptualization:** Young-Ki Lee.

**Data curation:** Do Hyoung Kim.

**Methodology:** Do Hyoung Kim.

**Validation:** Min-Kyung Kang, Seonghoon Choi.

**Writing – original draft:** Sun Ryoung Choi.

**Writing – review & editing:** Hayne Cho Park, Ajin Cho.

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
