## [Decision Letter · Decision Letter 0]

3 Mar 2021

PONE-D-20-35155

Central blood pressure and diastolic dysfunction predict CV mortality in dialysis patients with preserved systolic function

PLOS ONE

Dear Dr. Lee,

Thank you for submitting your manuscript to PLOS ONE. After careful consideration, we feel that it has merit but does not fully meet PLOS ONE’s publication criteria as it currently stands. Therefore, we invite you to submit a revised version of the manuscript that addresses the points raised during the review process.

We look forward to receiving your revised manuscript.

Kind regards,

Yoshihiro Fukumoto

Academic Editor

PLOS ONE

Journal Requirements:

2)  Please provide additional details regarding participant consent. In the ethics statement in the Methods and online submission information, please ensure that you have specified (1) whether consent was informed and (2) what type you obtained (for instance, written or verbal, and if verbal, how it was documented and witnessed). If your study included minors, state whether you obtained consent from parents or guardians. If the need for consent was waived by the ethics committee, please include this information.

3) Please include captions for your Supporting Information files at the end of your manuscript, and update any in-text citations to match accordingly. Please see our Supporting Information guidelines for more information: http://journals.plos.org/plosone/s/supporting-information.

4) Thank you for stating the following financial disclosure:

 [No].

Reviewers' comments:

Reviewer's Responses to Questions

**Comments to the Author**

1. Is the manuscript technically sound, and do the data support the conclusions?

Reviewer #1: Yes

Reviewer #2: Partly

2. Has the statistical analysis been performed appropriately and rigorously? 

Reviewer #1: Yes

Reviewer #2: I Don't Know

3. Have the authors made all data underlying the findings in their manuscript fully available?

Reviewer #1: No

Reviewer #2: Yes

4. Is the manuscript presented in an intelligible fashion and written in standard English?

Reviewer #1: Yes

Reviewer #2: Yes

5. Review Comments to the Author

Reviewer #1: I thoroughly enjoyed reading this report showing patients with LV diastolic dysfunction had

increased risk of CV mortality and that central systolic BP and diabetes were amongst the strongest predictors

of diastolic dysfunction. However, the lack of diagnostic information captured during the follow-up period slightly dampened my enthusiasm regarding the mechanism for CV mortality.

Despite their being no difference between clinical groups, I thought it was odd that BMI was not included as a predictor for LV diastolic dysfunction in the univarite and multivariate analyses given it being such and important risk factor in other CVDs, e.g. T2DM.

In the discussion, the authors did a good job summarizing extant literature that corroborates their findings, however there is no discussion pertaining to the next steps in this research. For example, the sample appears to not consist of individuals with elevated BMI. Would similar findings be expected?

Reviewer #2: This study was a single-center, prospective, observational study enrolled patients treated with HD or PD at Hallym University Kangnam Sacred Heart Hospital, which aimed to examine the impact of LV diastolic dysfunction on CV mortality in patients with ESKD with preserved systolic function and to analyze the relationship between central BP and LV diastolic dysfunction. In the present study, the authors revealed that LV diastolic dysfunction increased CV mortality in patients receiving HD and PD, in which E/E' ratio was a significant factor for CV mortality. Further, central systolic BP was a significant independent risk factor for LV diastolic dysfunction, and central systolic BP had a significant positive correlation with E/E’ ratio and LAVI. This reviewer considers that it was interesting, but that this paper has only a small impact. This reviewer has some criticisms as described below.

Major comments:

1. Title. The authors indicated that LV diastolic dysfunction, but not central systolic BP, could independently predict CV mortality in dialysis patients with preserved LV systolic function. And, central systolic BP was a significant independent risk factor for LV diastolic dysfunction. Therefore, the title “central blood pressure and diastolic dysfunction predict CV mortality” is wrong.

2. In the Results section, the authors indicated 16 deaths during the follow-up period, including 7 CV deaths and 9 other deaths, including pneumonia, peritonitis, and sepsis. In 7 patients, which of CV deaths occurred (heart failure, arrhythmia, etc.)?

3. Not only central systolic BP but also DM was a strong predictor of LV diastolic dysfunction. How did the authors consider this issue?

4. Discussion section, ref. 12 part. What was “Sumeet”?

5. Conclusion. It is known that LV diastolic dysfunction can predict CV mortality in dialysis patients. The combination of central systolic BP seems to be new, but it was not a predictor for CV mortality. How are the authors able to report new findings in the present study?

Minor comment:

6. Introduction. Line 5. Blank seems to be necessary between “[1].” and “The”.

7. Figure 1. “Left ventricular dysfunction” should be “Left diastolic dysfunction”.

6. PLOS authors have the option to publish the peer review history of their article (what does this mean?). If published, this will include your full peer review and any attached files.

Reviewer #1: No

Reviewer #2: No

---

## [Author Response · Author response to Decision Letter 0]

19 Mar 2021

5. Review Comments to the Author

Reviewer #1: I thoroughly enjoyed reading this report showing patients with LV diastolic dysfunction had increased risk of CV mortality and that central systolic BP and diabetes were amongst the strongest predictors of diastolic dysfunction. However, the lack of diagnostic information captured during the follow-up period slightly dampened my enthusiasm regarding the mechanism for CV mortality.

Despite their being no difference between clinical groups, I thought it was odd that BMI was not included as a predictor for LV diastolic dysfunction in the univarite and multivariate analyses given it being such and important risk factor in other CVDs, e.g. T2DM.

In the discussion, the authors did a good job summarizing extant literature that corroborates their findings, however there is no discussion pertaining to the next steps in this research. For example, the sample appears to not consist of individuals with elevated BMI. Would similar findings be expected?

Thank you very much for your kind suggestion. Tullio et al. reported that increased BMI (>25 kg/m2) was associated with LV diastolic dysfunction independent of LV mass and associated risk factors in general population (J Am Coll Cardiol 2011;57:1368–74). However, unlike the general population, lower BMI was related to mortality among hemodialysis patients (Hypertension.2011;58:1014-1020.). The exact reason for this interesting paradox is unknown. Meanwhile, Agarwal et al. showed an inverse relationship of BMI with unadjusted LVMI, but multivariate analysis removed the statistical significance. Unfortunately, Agarwal et al. did not perform further analysis to show relationship between BMI and LVDD. We could not find any studies on BMI and LVDD in dialysis patients. In our study, BMI was not a significant risk factor for LVDD in a univariate analysis (p=0.360). Furthermore, overweight and obesity with a BMI value of 25 or higher were not a significant risk factor for LVDD in a univariate analysis in this study (p=0.465). We believe that further studies on BMI, LVDD, and CV mortality in dialysis patients should be warranted and will be a great value. We have added the results of univariate analyses for BMI in Table 2 and Table 3. We have also added above description to the discussion section.

Reviewer #2: This study was a single-center, prospective, observational study enrolled patients treated with HD or PD at Hallym University Kangnam Sacred Heart Hospital, which aimed to examine the impact of LV diastolic dysfunction on CV mortality in patients with ESKD with preserved systolic function and to analyze the relationship between central BP and LV diastolic dysfunction. In the present study, the authors revealed that LV diastolic dysfunction increased CV mortality in patients receiving HD and PD, in which E/E' ratio was a significant factor for CV mortality. Further, central systolic BP was a significant independent risk factor for LV diastolic dysfunction, and central systolic BP had a significant positive correlation with E/E’ ratio and LAVI. This reviewer considers that it was interesting, but that this paper has only a small impact. This reviewer has some criticisms as described below.

Major comments:

1. Title. The authors indicated that LV diastolic dysfunction, but not central systolic BP, could independently predict CV mortality in dialysis patients with preserved LV systolic function. And, central systolic BP was a significant independent risk factor for LV diastolic dysfunction. Therefore, the title “central blood pressure and diastolic dysfunction predict CV mortality” is wrong.

Thank you for making an important comment. We sincerely apologize for the confusion. We have changed the title to “Clinical significance of central systolic blood pressure in LV diastolic dysfunction and CV mortality” to reflect the reviewer's comments. 

2. In the Results section, the authors indicated 16 deaths during the follow-up period, including 7 CV deaths and 9 other deaths, including pneumonia, peritonitis, and sepsis. In 7 patients, which of CV deaths occurred (heart failure, arrhythmia, etc.)?

Thank you for making an important point. Three out of seven died from myocardial infarction, one from sudden cardiac death, and the other three from stroke. We have added the above description in the result section.

3. Not only central systolic BP but also DM was a strong predictor of LV diastolic dysfunction. How did the authors consider this issue?

Thank you for making an important point. 

Diabetes is one of the most important metabolic conditions responsible for LVDD. Although the exact cause has not yet been identified, recently, diabetic cardiomyopathy has been proposed. Clinically, it is well known that LV diastolic function deteriorates with advancing age despite preserved LV systolic function. In particular, impaired glucose tolerance could contribute to the early deterioration of LV diastolic function among middle aged apparently healthy subjects. (Eur Heart J Cardiovasc Imaging 2020 Aug 1;21(8):885-886). Similarly, in this study, patients with diabetes had significantly more LVDD. Furthermore DM was identified as a very strong factor for LVDD with hazard ratio (HR) of 9.373 (p=0.001). The number of patients included in this study is small and cannot be generalized, but we think more research is needed. We have added what the reviewer points out to the discussion section.

4. Discussion section, ref. 12 part. What was “Sumeet”?

We are sorry for our mistake. There were some typos. We have amended the text as follows: Subherwal reported that central BP is independently associated with LV filling pressure and LV diastolic dysfunction in the general population [12].

5. Conclusion. It is known that LV diastolic dysfunction can predict CV mortality in dialysis patients. The combination of central systolic BP seems to be new, but it was not a predictor for CV mortality. How are the authors able to report new findings in the present study?

Thank you for making an important point. 

Although there are reports that hypertension affects CV mortality in dialysis patients (Hypertension. 2010;55:762–768, Hypertension. 2017;70:435–443) there are few studies on central BP and CV mortality in dialysis patients. We think that because the number of subjects was small and the observation period was not long enough, this study did not show significant results for central SBP on CV mortality. In addition, numerous risk factors involved in the pathogenesis of CVD in dialysis patients. This is likely due to cardiac dysfunction as well as non-traditional risk factors, such as chronic volume overload, anemia, inflammation, oxidative stress, chronic kidney disease–mineral bone disorder and other aspects of the ‘uraemic milieu’ (Nephrol Dial Transplant. 2018 Oct 1;33(suppl_3):iii28-iii34). Further research is likely to be needed. We have added what the reviewer points out to the discussion section.

Minor comment:

6. Introduction. Line 5. Blank seems to be necessary between “[1].” and “The”.

Thank you for the important point. We have amended the text as you have recommended.

7. Figure 1. “Left ventricular dysfunction” should be “Left diastolic dysfunction”.

Thank you for the important point. I modified it as reviewer comment.

---

## [Decision Letter · Decision Letter 1]

12 Apr 2021

Clinical significance of central systolic blood pressure in LV diastolic dysfunction and CV mortality

PONE-D-20-35155R1

Dear Dr. Lee,

We’re pleased to inform you that your manuscript has been judged scientifically suitable for publication and will be formally accepted for publication once it meets all outstanding technical requirements.

Kind regards,

Yoshihiro Fukumoto

Academic Editor

PLOS ONE

Additional Editor Comments (optional):

Reviewers' comments:

Reviewer's Responses to Questions

**Comments to the Author**

1. If the authors have adequately addressed your comments raised in a previous round of review and you feel that this manuscript is now acceptable for publication, you may indicate that here to bypass the “Comments to the Author” section, enter your conflict of interest statement in the “Confidential to Editor” section, and submit your "Accept" recommendation.

Reviewer #2: All comments have been addressed

2. Is the manuscript technically sound, and do the data support the conclusions?

Reviewer #2: Yes

3. Has the statistical analysis been performed appropriately and rigorously? 

Reviewer #2: I Don't Know

4. Have the authors made all data underlying the findings in their manuscript fully available?

Reviewer #2: Yes

5. Is the manuscript presented in an intelligible fashion and written in standard English?

Reviewer #2: Yes

6. Review Comments to the Author

Reviewer #2: This reviewer considers that the authors have well responded. This reviewer has no further comment.

7. PLOS authors have the option to publish the peer review history of their article (what does this mean?). If published, this will include your full peer review and any attached files.

Reviewer #2: No

---

## [Editor Report · Acceptance letter]

14 Apr 2021

PONE-D-20-35155R1 

Clinical significance of central systolic blood pressure in LV diastolic dysfunction and CV mortality 

Dear Dr. Lee:

I'm pleased to inform you that your manuscript has been deemed suitable for publication in PLOS ONE. Congratulations! Your manuscript is now with our production department. 

Kind regards, 

on behalf of

Dr. Yoshihiro Fukumoto 

Academic Editor

PLOS ONE